Acidity, sugar, and alcohol contents during the fermentation of Osmanthus-flavored sweet rice wine and microbial community dynamics

Tian Ping 1
Wan Jiaqiong 1
Yin Tuo 1
Liu Li 2
Ren Hongbing 2
Cai Hanbing 1
Liu Xiaozhen 1
http://orcid.org/0000-0001-7440-4348 Zhang Hanyao 1 zhanghanyao@swfu.edu.cn
1 Key Laboratory for Forest Resources Conservation and Utilization in the Southwest Mountains of China, Ministry of Education, Southwest Forestry University , Kunming , China
2 R&D Department, Honghe Hongbin Food Co., Ltd. , Jianshui , China
Okpala Charles
Electronic publication date: 2025 Jan 30
Publication date: 2025
Volume: 13
Electronic Location ID: e18826
Received 2024 Jul 18; Accepted 2024 Dec 17
Copyright: © 2025 Tian et al.
Copyright year: 2025
Copyright holder: Tian et al.
License: This is an open access article distributed under the terms of the Creative Commons Attribution License, which permits unrestricted use, distribution, reproduction and adaptation in any medium and for any purpose provided that it is properly attributed. For attribution, the original author(s), title, publication source (PeerJ) and either DOI or URL of the article must be cited.
License URL: https://creativecommons.org/licenses/by/4.0/

Keywords: Sweet rice wine, Osmanthus flower, High-throughput sequencing, Microbial communities and differences, Dominant strains

Funding: Major Scientific and Technological Project in Yunnan Province-Biological Seed Industry and Agricultural Products Deep Processing Project 202302AE090019 National Natural Science Foundation of China 31760450 Yunnan First-class Construction Discipline of Forestry Science of Southwest Forestry University This study is supported by a Major Scientific and Technological Project in Yunnan Province-Biological Seed Industry and Agricultural Products Deep Processing Project (202302AE090019), the National Natural Science Foundation of China (Grant No. 31760450), and the Yunnan First-class Construction Discipline of Forestry Science of Southwest Forestry University. The funders had no role in study design, data collection and analysis, decision to publish, or preparation of the manuscript.

==============================
Sweet rice wine is a popular traditional Chinese rice wine widely loved by Chinese people for its high nutritional value. Osmanthus flower petals contain various nutrients and have good medicinal value. However, the dynamics of the sugar level, acidity, alcohol content, and microbial community during the fermentation of Osmanthus-flavored sweet rice wine have not been evaluated, which can lead to the unstable quality of Osmanthus flower sweet rice wine (OFSRW). In this study, the dynamic changes in sugar level, acidity, alcohol content, microbial community composition, and microbial metabolic pathways during traditional fermentation of OFSRW at four-time points—0 h (AG0), 24 h (AG24), 36 h (AG36), and 43 h (AG43)—were analyzed via direct titration, total acid assays, alcoholometry, and high-throughput macrogenomic techniques. First, we found that bacteria were the dominant microorganisms in the early stage of OFSRW fermentation (AG0), fungi were the dominant microorganisms in the middle and late stages of fermentation (AG24 and AG36), and Rhizopus was the main fungal genus throughout fermentation. Acidity and total sugars increased with fermentation time, and alcohol was not detected until the end of fermentation. Diversity analysis revealed that the dominant species at the beginning of natural fermentation was A. johnsonii, and R. delemar became the dominant species as natural fermentation progressed. Metabolic pathway analysis revealed that energy production and conversion, carbohydrate transport, amino acid transport, and metabolic pathways were the most active metabolic pathways in the fermenter. These results provide a reference basis for changes in the microbial community during the fermentation of cinnamon-flavored sweet rice wine.

Introduction

Osmanthus flower sweet rice wine (OFSRW), a low-alcoholic beverage with regional characteristics in China, is made mainly from high-quality glutinous rice and Osmanthus flowers, which are mixed and fermented in a natural environment with the addition of a fermentation agent (Jiuqu). Many studies have confirmed that glutinous rice is a complex organism composed of various macromolecules with edible and medicinal value (Zheng et al., 2023). For example, it warms the spleen and stomach, stops cold dysentery deficiency, reduces stool, allows spontaneous sweating, and facilitates urination. Osmanthus fragrans (Thunb), an evergreen shrub or tree belonging to the Lignaceae (Oleaceae) family, is a valuable and common ornamental aromatic plant with good medicinal value (Huang et al., 2019b). Its petals contain many nutrients, such as soluble sugars, soluble proteins, organic acids, vitamin C, flavonoids, free amino acids, and many minerals. It has the effects of strengthening the stomach and resolving phlegm, generating fluids, dispersing blood stasis, and flattening the stomach (Zhou & Yue, 2013). It can treat asthma, cough, toothache, and diarrhea (Wang et al., 2022), as well as have anticancer, antioxidant, and anti-inflammatory effects (Huang et al., 2023). Sweet rice wine is smooth and long-lasting, has a pleasant aroma, and is an essential part of the food culture of Chinese people. The nutritional value of rice wine is also noteworthy. In addition to the nutritional components of the raw materials, various components, such as oligosaccharides, peptides, proteins, B vitamins, minerals, and amino acids, are also produced during traditional brewing. These components are easily digested and absorbed by the human body, promoting appetite, warming the stomach and strengthening the spleen, benefiting qi and preventing diarrhea, generating fluids, stopping sweating, invigorating the spirit, and relieving fatigue (Yuan, Zhang & Fang, 2020; Cai et al., 2012), and exerting antiage effects (Liu et al., 2020a; Zhao et al., 2018). Therefore, the OFSRW produced by combining glutinous rice and osmanthus flowers through the mixed fermentation of wine curd has the aroma of ordinary sweet rice wine and retains the fresh fragrance of Osmanthus flowers, which not only increases the color, aromas, and taste of sweet rice wine but also allows the release of nutrients such as oligosaccharides, polypeptides, amino acids, and ethanol in sweet rice wine to increase its nutritional value.

Jiuqu is rich in microorganisms involved in saccharification, fermentation, and the production of flavor-related metabolites. For example, Rhizopus breaks the α-1,4 and α-1,6 bonds in the rice starch structure, converted more completely into fermentable sugars. Saccharomyces plants draw monosaccharides such as glucose, fructose, and mannose into the cell and break them down into carbon dioxide and ethanol under anaerobic conditions through the action of endoenzymes. Lactic acid bacteria (LAB) play a prime role in fermentation and the intrinsic properties of fermented products, influencing the development of their aroma, texture, and acidity (Cai et al., 2019). The fungal communities in Jiuqu have also been shown to play essential roles in starch and protein hydrolysis and the production of ethanol, organic acids, higher alcohols, and esters (Medina et al., 2013). Bacterial communities that produce hydrolytic enzymes, glucoamylases, proteases, and esterases are produced by various bacteria during fermentation to degrade the raw material substrates, all of which could lead to the accumulation of aroma-related compounds or secondary metabolites and intermediates (Gammacurta et al., 2018; Simonen & Palva, 1993). The bacterial and fungal communities varied significantly among the different starters of Hongqu yellow rice wine, and the core microorganisms were positively correlated with specific organic acids and aromatic esters in the starters (Huang et al., 2019a). Regional variations in wild native microbes and the surrounding conditions of Jiuqu production may influence the microbial community composition and quality of Jiuqu, especially in nonsterile fermentation processes (Zhao et al., 2022a). The influence of different regional environments, raw materials, and additives in sweet rice wines may also lead to differences in sweet rice wine (SRW) fermenters, which may confer different organoleptic characteristics, flavors, and other features of SRW (Su & Zhao, 2014). For example, Chen et al. (2020) reported that different microbial communities in three different traditional huangjiu fermenters resulted in significant differences in the aroma composition of their fermented rice wines. One study comparing eight Chinese sweet rice wine (CSRW) starter samples from different regions of southern China revealed significantly high variation in bacterial and fungal composition, which likely contributed substantially to the final flavor quality of the respective CSRW (Cai et al., 2018). Sugars, acidity, and alcohol content, which are crucial parameters in the sweet rice wine fermentation process, strongly affect the quality and flavor of sweet rice wine. For example, insufficient sweetness or oversweetness makes sweet rice wine taste too light or too mushy. Excessive alcohol makes sweet rice wine bitter and astringent, with a slight off-flavor, a strong taste, or even a distinct white wine taste. Excessive acidity can reduce the taste of sweet rice wine or even cause rancidity (Wu et al., 2016).

OFSRW fermentation is a complex process involving maceration, rinsing, steaming, fermentation inoculation, addition of Osmanthus flowers and saccharification. The entire fermentation process includes a range of strains obtained from fermenters, raw materials, and the environment (He, Lin & Tong, 2015; Zhao et al., 2022a). In addition, in this complex environment, a series of changes in sugar level, acidity, alcohol content, and microflora occur, affecting the unique aroma, flavor, and color of rice wine. Furthermore, OFSRW fermentation follows the traditional technique of an uncontrolled fermentation process that produces inconsistent flavors. However, few studies have evaluated the evolution of sugar, acidity, alcohol, and microbiota during traditional OFSRW fermentation, and the interactions between microbiota and sugar, acidity, and alcohol have not been elucidated. To explain this, it is first necessary to understand the dynamic relationships among the brix, acidity, alcohol, and microorganisms in OFSRW. In this work, direct titration was used to determine the sugar level in OFSRW. Total acidity determination was used to determine the acidity of the OFSRW, and alcoholic strength was used to determine the alcoholic strength of the OFSRW. Available methods, including culture-dependent methods (Lv et al., 2012) and culture-independent PCR-denaturing gradient gel electrophoresis (DGGE) techniques (Lv et al., 2015, 2017), have been employed to study the microbial composition of OFSRW, but all of the above methods have difficulties distinguishing the species present at population densities below 103 CFU/g or two orders of magnitude lower than the most abundant members of these communities (Cocolin et al., 2011; Prakitchaiwattana, Fleet & Heard, 2004). High-throughput sequencing technology, on the other hand, is capable of analyzing the transcriptome and genome data of a species in a detailed and comprehensive manner, also known as deep sequencing or next-generation sequencing (NGS). This technique has quantitative capabilities for determining the abundance of species components in a sample. In addition, the utilization of this technique is simple and cost-effective, the results are feasible (Liu et al., 2019), and it is faster and better than ITS PCR and fluorescent ITS PCR capillary electrophoresis. This technique has been widely used to analyze the microbial community dynamics of various fermented foods and vegetables, such as Sichuan kimchi (Luo et al., 2021), soy sauce (Zhao et al., 2021), kiwifruit (Zhang et al., 2022a), grape juice (Zhao et al., 2022b), and rice wine (Zou et al., 2023), and the use of macro-genome sequencing in these studies provided a theoretical basis for analyzing the relationships between microbial populations and specific flavors in these fermented foods.

High-throughput macrogenomic rDNA sequencing technology, with merits such as low cost, high feasibility, and wide application, coupled with the fact that it has not yet been applied to the analysis of microbial community dynamics during the fermentation process of osmanthus-flavored sweet rice wine (OFSRW). Therefore, in this article, the microbial community dynamics during the traditional fermentation of osmanthus-flavoured sweet rice wine (OFSRW) were monitored using high-throughput macrogenomic rDNA sequencing: (1) Changes in the brix, acidity, alcohol content, and microbial community composition of OFSRW were analyzed. (2) Comparisons were made between the OFSRW to determine the alcoholic strength of OFSRW microbial diversity changes and differences during fermentation and the correlations between the microbial community and sugar level, acidity, and alcohol content. (3) The functional metabolism prediction of OFSRW. This study provides insights into how microorganisms in OFSRW fermentation broth adapt to environmental changes during fermentation and can also be crucial for optimizing fermentation conditions and improving product quality and flavor.

Materials and Methods

Sample preparation and sampling

This study used high-quality glutinous rice and dried Osmanthus flowers as raw materials for sweet rice wine production and conducted single-factor experiments. Measurements include the product’s sugar content, acidity, alcohol content, and changes in microbial communities. The raw materials used for OFSRW were ‘Angel Rice Leaven’ (Hubei Angel Yeast Co., Ltd., Hubei, China), glutinous rice, and osmanthus (Guangzhou Zhenyuantang Food Co., Ltd., Guangzhou, China). Angqi sweet wine yeast is a commercial yeast product. The OFSRW production process is shown in Fig. 1. In the production of OFSRW, it is necessary to select glutinous rice that is full of grains, has no yellowing and no mold, and has no insect pests. When making OFSRW, glutinous rice with whole grains, no yellowing, no mildew, and no insect pests were selected, and the rice was washed into a pottery jar with 2.5–3.0 times the quality of glutinous rice to soak for 9–12 h. The glutinous rice soaked at 108 °C was steamed for 40–50 min, and the steamed rice was hard on the outside and soft inside, fluffy and not rotten, with no white heart or uniformity. After the glutinous rice was steamed and cooked, 30% cool white boiled water was used to cool it to 30–35 °C, after which 0.4% Jiuqu (Li et al., 2018) and rice were added and stirred evenly, and 0.75% dried cinnamon was added again and stirred well. Finally, the mixture was bottled, compacted, mashed, and placed in a constant temperature box at 30 °C for 43 h of fermentation. At 0, 24, 36, and 43 h, 20 g samples were randomly taken to determine sugar, acidity, and alcohol content. At the same time, 380 mL of fermentation mixture was collected for high-throughput sequencing analysis. When collecting samples, the samples from different stages were placed on a super-clean workbench, and sterilized equipment was used for sampling, ensuring that the samples were not recontaminated by the external environment during the process.

Figure 1 Flow chart for making Osmanthus-flavored sweet rice wine.

Determination of brix, acidity, and alcohol content

The direct titration method GB 5009.7-2016 was used to determine the sugar content in Osmanthus sweet rice wine (National Health Commission of the People’s Republic of China, 2016). OFSRW was sent to the Laboratory Department of Hongbin Foodstuffs Co. Ltd. to test its acidity, and the method of determination was GB 12456-2021 Total Acid Determination Method (National Health Commission of the People’s Republic of China, 2021). The alcohol content was determined via the GB 5009.225-2023 alcoholometer method (National Health Commission of the People’s Republic of China, 2023). The means and standard deviations were analyzed via the SPSS software package.

High-throughput sequencing and bioinformatics analysis of extracted and sequenced DNA

Total DNA was extracted from each microbiological sample using a DNA extraction kit. The total DNA extraction process is shown in Fig. 2. High-throughput sequencing was performed by DynaTech Biotechnology Limited (Yunnan Province, China) for macro-genome sequencing. The sample DNA fragments were paired-end sequenced with the Illumina HiSeq X platform. Metagenomic sequencing was conducted, and ITS5 (GGAAGTAAAAGTCGTAACAAGG) and ITS2 (GCTGCGTTCTTCATCGATGC) were used for sequencing (Kumar, Eble & Gaither, 2020). The sequences had a mean read length of 150 bases and a Q score of 30. The sequencing data were analyzed via the GenesCloud platform (www.genescloud.cn).

Figure 2 Flowchart of total DNA extraction.

Sequence processing and analysis

Paired-end sequencing of DNA fragments was performed on the Illumina platform. Vsearch (v2.13.4-linux-x86_64) and cutadapt (v2.3) were used to denoise and cluster the sequences (Rognes et al., 2016), whose global paired-end sequence comparison is a core function of Vsearch. After using the QIIME cutadapt trim paired to excise sequence primer fragments, the sequences unmatched with primers were discarded.

The Vsearch module was used for splicing, deduplicating, and dechimericing the sequences, and the UNITE database (Release 8.0, https://unite.ut.ee/) was used to filter the concentrated chimeras to obtain high-quality chimeras. QIIME2 (classify sklearn algorithm https://github.com/QIIME2/q2-feature-classifier) was used to annotate the characteristic sequences of each operational taxonomic unit (OTU) in the naive Bayes classifiers (Elolimy et al., 2020). The QIIME2 qiime feature-table Rarefy function was used to set the leveling depth to 95% of the smallest sample sequence size, and the final OTUs were obtained.

Analysis of changes in species composition

Krona software (https://github.com/marbl/Krona/wiki) was used to analyze the community taxonomic composition of the samples (Ondov, Bergman & Phillippy, 2011), which is centered on the use of multi-layered pie charts to explore stratified data interactively. The RGGplot2 package was used to construct a circle stair tree diagram, and the abundance of each group was added to the diagram in the form of a pie chart (Steenwyk & Rokas, 2021). To further compare the species composition differences among the samples and display the distribution trend of the species abundance of each sample, R language and the pheatmap package were used to construct heatmaps for the species composition analysis.

Alpha diversity analysis

Alpha diversity refers to the diversity within a sample and focuses on the degree of microbial diversity in the sample, including species richness and evenness of distribution of individuals in the community. Commonly used alpha diversity indices include the Chao1 estimator, Good’s coverage index, the observed species index, Pielou’s evenness index, the Shannon index, the Simpson index, etc. (Chao & Ricotta, 2019; Liu et al., 2020b). The Chao 1 index measures species richness and estimates the number of species in a sample. Good coverage refers to the coverage rate of each sample library, and this index reflects whether the sequencing result represents the situation of the microorganisms in the sample. The observed species index represents the number of species in the samples. Pielou’s evenness indicates the uniformity of the community. The Shannon index combines abundance and evenness, giving more weight to rare species. The Simpson index combines abundance and evenness but focuses more on common species.

QIIME2, R language, and the ggplot2 package were used for alpha diversity analysis. After the unleveled OTU table was used, the “qiime diversity alpha-rarefaction” command was used, the minimum leveling depth was set to ten, and the minimum sequencing depth was set to 95% of the sample sequence. Each depth value was fattened ten times to calculate the alpha diversity index.

Beta diversity content analysis

Beta diversity refers to the differences between samples or groups and is often used to analyze whether the differences in microbial composition between two groups are significant. Commonly used beta diversity indices include the Jaccard, Bray‒Curtis, unweighted UniFrac, and weighted UniFrac indices (Lozupone et al., 2007; Chao & Ricotta, 2019). The Jaccard index compares the similarities and differences between limited sample sets. Bray–Curtis dissimilarity is a measure used to analyze differences in species composition across different places. Unweighted UniFrac can detect the presence of variations between samples, whereas weighted UniFrac can further quantitatively detect the variation between samples of various lineages.

Principal coordinate analysis (PCoA) and nonmetric multidimensional scaling (NMDS) methods are used to analyze the beta diversity of the samples (Ramette, 2007; Legendre & Legendre, 1998) in order to look at the differences in the structure of the microbial communities between the samples, where the closer the samples were to each other, the more similar the species composition was, and vice-versa for more distant samples. By default, the UPGMA algorithm was used for cluster analysis of the Bray‒Curtis distance matrix (Bray & Curtis, 1957), and a ggtree of R language was used to analyze the relationships between different samples for visualization.

Analysis of functional prediction

Functional annotation information was obtained by comparing the nonredundant genes with each functional database via DIAMOND software, taking the annotations with e < 1e−5, and filtering the proteins with the most abundant sequences. For each sequence comparison result, the comparison result with the highest SCORE (oneHSP > 60 bits) was selected for subsequent analysis (Backhed et al., 2015). The abundance values of metabolic pathways were obtained via PICRUSt2 software (https://github.com/picrust/picrust2/wiki). The generated data were entered into the KEGG biological metabolic pathway analysis database (KEGG Pathway Database, http://www.genome.jp/kegg/pathway.html), the eggNOG database, and metabolic pathways for different sample statistics (Qin et al., 2012). Using R language and the MetaGenomeseq package, the Fit Feature Model function was employed, and the distribution of each pathway/group was analyzed via a zero-log-normal model. The results were used to calculate the significance of the metabolic differences between each natural fermentation sample and the CK control group via the Kruskal‒Wallis rank sum test to determine whether there was a difference in the derived function in the data. According to the data selected in the metabolic pathway abundance table, a bar chart was drawn to use R language to analyze which species affect the metabolic pathways.

Statistical analysis

Mean and standard deviation analyses were performed using SPSS. The richness and evenness of microorganisms in the samples were counted using alpha diversity analysis. Comparison of microbial community structure between samples at different periods was analyzed using principal coordinate analysis (PCoA) and non-metric multidimensional scaling (NMDS) methods based on weighted and unweighted homogeneous distances. The significance of metabolic differences between each natural fermentation sample and the CK control was calculated using the Kruskal-Wallis rank sum test.

Results

Brix, acidity, and alcohol content analysis

During OFSRW fermentation, the time before fermentation (AG0h) was 0 h, the brix of OFSRW was undetectable, the acidity of OFSRW was 0.03, and the alcoholic strength of OFSRW was undetectable, indicating that fermentation had not yet begun, that no fermentable sugar existed in the raw material and that the acidity was low, which might be related to the activity of L. plantarum. In the middle of fermentation (AG24h and AG36h), the sugar level of AG24h was 7.11, the acidity was 0.14, and the alcoholic strength was not detected. However, the sugar level of AG36h was 14.05, the acidity was 0.25, and the alcoholic strength was not detected. The increase in brix and acidity during the middle fermentation period also confirms the association of enhanced microbial activity, especially related to the activity of LAB and Rhizopus, which can produce acid through the metabolism of sugars during fermentation. An increase in acidity helps prevent the growth of other microorganisms, thus controlling the structure of the microbial community to some extent during fermentation, and an increase in acidity decreases the viability of LAB. The sugar level after fermentation (AG43h) was 16.55, the acidity was 0.35, and the alcohol content was 2%Vol, which indicated that the fermentation process had begun to enter the alcoholic fermentation stage (Table 1). The fact that alcohol was not detected until the end of fermentation may be due to the inhibition of Saccharomyces and Rhizopus activity by L. plantarum during the pre-fermentation period (Deng, Du & Xu, 2020; Yao et al., 2019; Wang et al., 2023; Liang et al., 2022; Gupta & Srivastava, 2014).

Table 1 Physicochemical properties of sweet rice wine produced from Cinnamon osmanthus flowers.

Local flavor fermentation cycle	OFSRW	
Physical and chemical indicators	0 h	24 h	36 h	43 h	
Acidity	0.03 ± 0.001	0.14 ± 0.010	0.25 ± 0.015	0.35 ± 0.015	
Total sugar (g/250 ml)	Not detected	7.11 ± 0.026	14.15 ± 0.21	16.53 ± 0.07	
Alcoholic strength (v/vol, %)	Not detected	Not detected	Not detected	1.96 ± 0.05	

Results of the species composition

In this study, the dynamics of fungal and bacterial communities in OFSRW fermenters at different time intervals were detected via Krona analysis (Fig. 3). According to the results of the present study, in the AG0 samples, bacteria were dominated by Gammaproteobacteria and Bacilli, accounting for 76% and 18%, respectively. Fungi were dominated by Mucoromycetes and Magnoliopsida at 51% and 40%, respectively. For the AG24 sample, the bacteria were the same as those in the AG0 stage. The fungi were dominated by Saccharomycetes and Mucoromycetes at 86% and 8%, respectively. In the AG36 samples, the number of fungi decreased, and the number of bacteria increased, but fungi remained the dominant species. The bacteria were dominated by 11% Enterobacterales. Fungi were dominated by Mucorales and Saccharomycetales at 66% and 15%, respectively. In the AG43 samples, Mucorales was the dominant order of fungi, accounting for 83% of the fungi. The bacteria were dominated by 6% and 3% Enterobacterales and Moraxellales, respectively.

Figure 3 Classification level and abundance information of Krona diagrams of sample species.

From the inside to the outside, the Krona circle represents the seven taxonomic levels of domain, phylum, class, order, family, genus, and species. The sector size reflects the relative abundance of different taxa, and there are specific values.

The 30 most abundant species for each sample were plotted as a bar graph (Fig. 4). The dominant species in AG0 were A. johnsonii (abundance value = 25,328.43406), P. pentosaceus (abundance value = 19,416.53625) and L. plantarum (abundance value = 13,204.09352). The significant activity of P. pentosaceus and L. plantarum also explains the low acidity at this stage. However, as fermentation continued, A. johnsonii and P. pentosaceus disappeared. After 24 h of fermentation, the K. ascorbata content decreased. The dominant fungi were C. lusitaniae (abundance = 347,248.9604) and R. delemar (abundance = 20,415.18217). The abundance of bacteria subsequently increased, but fungi remained dominant, and the abundance of C. lusitaniae decreased or disappeared. The significant activity of Rhizopus indicates the beginning of fermentation of starch in rice, thus converting it into fermentable sugars. C. lusitaniae is a non-Saccharomyces yeast species capable of utilizing sugars for fermentation, resulting in turbidity and precipitation of OFSRW and alcohol production (Cao et al., 2014). After 36 h of fermentation, Rhizopus accounted for the greatest percentage, with R. delemar (abundance = 221,286.5492) being the dominant strain, followed by R. microsporus (abundance = 42,293.8848). W. anomalus (abundance = 74,557.64045) was dominant among the yeasts. The continuous activity of Rhizopus at this stage indicates an inextricable relationship with the sustained increase in Brix. W. anomalus is a non-Saccharomyces yeast with certain aroma-producing, ester-producing, and alcohol-producing abilities that can significantly increase the sensory quality of wine, and it is an important functional microorganism in the fermentation of wine grains (Xie et al., 2022). The significant activity of Rhizopus, C. lusitaniae and W. anomalus in the middle and late stages of fermentation paved the way for monitoring the alcohol content in the alcoholic fermentation stage (AG43). As fermentation continued, the yeasts decreased or disappeared. At 43 h of fermentation, Rhizopus became dominant, especially R. delemar (abundance value = 284,014.9015), which became the dominant strain, and R. microsporus (abundance value = 52,561.42163) accounted for a greater percentage. At this stage, the alcohol content was tested at 2% Vol, but Saccharomyces decreased or disappeared, indicating that the elevated alcohol content inhibited yeast activity. However, Rhizopus was still the dominant fungus; the continuous increase in both the detected alcohol content and the sugar content was related to Rhizopus because Rhizopus has abundant amylase and certain liquefaction enzymes, which can chain OFSRW with saccharification and fermentation throughout the whole fermentation process from the beginning to the end of the fermentation process, and the fermentation effect was more thorough; thus, the starch yield further improved (Liu, 2017; Long et al., 2013; Nout & Aidoo, 2002; Crabb & Shetty, 1999).

Figure 4 Column diagram of the horizontal species composition of each sample species.

The abscissa is the name of each group of the grouping scheme, and the ordinate is the relative abundance of each taxon at a specific taxonomic level. Different colors represent different strains of bacteria.

Heatmaps were generated based on the average abundance of the top 50 strains, reflecting the correlation of colonies between samples and showing the trend of the distribution of strains in each sample. The results are shown in Fig. 5. The relationship between each sample and each colony can be seen in the heatmap. The species diversity was highest at AG36. The diversity gradually increased as fermentation progressed. The results of the heatmap analysis were consistent with the results of the species composition analysis.

Figure 5 Horizontal distribution heatmap of each sample species.

The x-axis represents sample groups, the left y-axis panel shows the hierarchical clustering analysis of selected samples, and the right panel shows the names of 50 bacterial species.

Analysis of the alpha diversity index

The Shannon, Simpson, and invsimpson indices were calculated to characterize the alpha diversity of the microbiota in each of the starting samples (Fig. 6). The Shannon and Simpson indices reflect the diversity of the microbial community, with higher Shannon scores and lower Simpson scores indicating greater diversity of the microbial community. There are differences between bacteria and fungi. During OFSRW fermentation, the highest species diversity was detected in sample AG0, and the lowest diversity was detected in AG24. In the four fermentation broth samples of AG0 (CK), AG24, AG36, and AG43, the Shannon index first decreased but then increased and then decreased. The Simpson index also showed the same trend as the Shannon index, which indicated that the species diversity first decreased and then increased and then decreased with the continuation of natural fermentation.

Figure 6 Alpha diversity indices among the samples.

The x-axis represents the sample groups, and the y-axis represents the values of the diversity analysis indices for the selected samples. Different colored dots indicate different analysis metrics.

Beta diversity analysis

The results of the beta diversity analysis are shown in Fig. 7. When AG0 was used as the sample control (CK), the species composition of the AG0 samples was far from the species composition of the AG24, AG36, and AG43 samples. The difference in species composition was large, which indicated that the number of strains in the CK samples was not large and that the colony structure was more different from that in the AG24, AG36, and AG43 samples.

Figure 7 PCoA and NMDS diagram.

Each dot in the figure represents a sample, and different colored dots indicate different samples (groups).

The similarity between the samples is shown in the form of a hierarchical tree (Fig. 8), with the AG36 and AG43 samples having the closest species composition distances, suggesting that the two samples are most similar. Taken together, these results indicate that the microbial communities of OFSRW fermenters from different periods were variable and similar. The clustered hierarchical tree revealed that the proportion of R. delemar was greater in samples AG36 and AG43, but the largest proportion of C. lusitaniae was found in sample AG24. On the other hand, the species composition of sample AG0 was the farthest from that of the other samples, indicating that the species composition of AG0 was different from that of the other samples. The difference between them was significant. This result is consistent with the results of the PCoA and NMDS analyses.

Figure 8 Hierarchical cluster analysis among samples.

The upper panel shows a hierarchical clustering tree diagram in which the samples were clustered according to similarity. The shorter the branch length between the samples was, the more similar the samples were. The lower panel is a stacked histogram of the 30 most abundant species.

Prediction of microbial function

The functional metabolic capacity of the microbial community was inferred from the composition of 16S rRNA genes in the macrogenomic data of different fermenters (Chen et al., 2020). Among the first-order KEGG metabolic pathways, the predicted functional genes enriched in the OFSRW fermenters were related to cellular processes, environmental information processing, genetic information processing, human diseases, metabolism, and organismal systems. A comparison of the abundances of the four samples is shown in Fig. 9. Metabolism-related pathways were significantly enriched in most of the samples, especially AG0, whereas the abundance of predicted genes related to organismal systems was relatively low in the AG0 samples; however, the metabolic pathways declined and then stabilized with increasing fermentation time, whereas the number of organismal systems metabolic pathways was greater than that in the AG0 fermentation samples in AG24, AG36, and AG43.

Figure 9 Comparison of KEGG primary metabolic pathways.

The vertical coordinates are the mean values of the abundance of functional pathways in the selected samples, and the horizontal coordinates are the sample groupings. Different colors represent different metabolic pathways.

To investigate the reasons for the changes in functional pathways, we conducted statistics on the secondary pathways involved in OFSRW metabolism, which encompass 45 secondary metabolic pathways. See Table S1 for details. The secondary metabolic pathways associated with increased expression were subjected to heatmap analysis, as shown in Fig. 10. The most abundant predicted metabolism in the category of level 2 KEGG pathways was energy production and conversion, followed by inorganic ion transport and metabolism, carbohydrate transport, amino acid transport and metabolism, nucleotide transport and metabolism, and lipid transport and metabolism. Inferred carbohydrate transport, amino acid transport, and metabolism were prominent in sample AG0; energy generation and conversion metabolic pathways were prominent in sample AG24; posttranslational modification, protein turnover, and chaperone metabolic pathways were prominent in sample AG36; and chromatin structure and dynamics were prominent in sample AG43, which may have contributed to the observed variations in volatile compound profiles between these samples.

Figure 10 Heatmap of the horizontal distribution of KEGG secondary metabolic pathways for each sample.

The x-axis represents sample groups, the left panel shows the hierarchical clustering analysis of the selected samples, and the right panel displays the names of metabolic pathways.

Discussion

Brix, acidity, and alcohol content are the most important parameters used to monitor the fermentation process of Osmanthus flower sweet rice wine. Glucose, sucrose, and maltose are three major fermentable sugars in the fermentation process of yellow rice wine because the starch in rice/wheat is predominantly degraded by α-amylase and glucosidase from Jiuqu (Kim & Seo, 2021; Yu et al., 2015). During fermentation, some low-molecular-weight sugars are consumed by microorganisms, contributing to an increase in the organic acid content (Huang et al., 2019a). In this study, the time before fermentation (AG0h) was 0 h, the sugar content was not detected, the acidity was 0.03, and the alcohol content was not detected (refer to Table 1), which indicated that fermentation had not yet begun, that essentially, no fermentable sugar existed in the raw material and that the acidity was low, which might be related to the activity of L. plantarum. The sugar content and acidity gradually increased with fermentation time. This may be related to the gradual decomposition of other sugars and indicates that the microorganisms became active and consumed sugars to produce energy. The gradual increase in acidity (from 0.03 to 0.35) also confirms the intensification of microbial activity (refer to Table 1), especially the activity of acid-producing bacteria such as P. pentosaceus and L. plantarum (refer to Fig. 4), which are capable of producing acid through the metabolism of sugars during fermentation. The sustained increase in acidity is consistent with previous findings (Tian et al., 2022). The increase in acidity may be related to the decrease in L. plantarum, which lowers the pH of OFSRW and inhibits the growth of spoilage microorganisms that are sensitive to acidic conditions (Perpetuini et al., 2020), thereby controlling, to some extent, the fermentation microbial community structure during the process. Rhizopus, as the main fungal genus throughout the fermentation process, could more completely convert the starch in glutinous rice into fermentable sugars, indicating that the presence of Rhizopus was inextricably linked to the increase in sugar content.

In this study, by the end stage of fermentation (AG43h), the first brix of 2% Vol was detected (refer to Table 1), which indicated that the fermentation process had started to enter the alcoholic fermentation stage. The lack of alcohol detection in the first three fermentation stages could also be related to L. plantarum since LAB has strong inhibitory activity against E. coli, Saccharomyces, and Mucor under low pH conditions (Russo et al., 2017). Alcohol production marks the beginning of the conversion of sugars into alcohol and carbon dioxide by Saccharomyces, which is typical of the traditional alcoholic fermentation phase (Fugelsang & Edwards, 2007; Fleet, 1993). The detection of alcohol at the end stage of fermentation (AG43h) may be related to the significant activity of Saccharomyces and Rhizopus in the middle and late stages of fermentation (AG24 and AG36), where Rhizopus is capable of hydrolyzing starch to obtain sugars. Saccharomyces is capable of fermenting sugars to produce alcohol, the especially notable activity of C. lusitaniae and W. anomalus (refer to Fig. 4), because C. lusitaniae can utilize sugars in the fermentation of alcohol, and W. anomalus can secrete a variety of glycosidases, such as β-D-glucosidase, β-D-xylosidase, and α-L-rhamnosidase, which can promote the formation of aroma and flavor substances and can produce high amounts of ethyl acetate and 2-phenylethanol, which can significantly improve the sensory quality of the wine body (Padilla, Gil & Manzanares, 2018; Sabel et al., 2014; Sun et al., 2022). However, the trend of the alcohol content in this study was opposite to that of the total sugar content (refer to Table 1), possibly because sugar has not yet been converted to ethanol and L. plantarum during the continuous fermentation process.

Microorganisms play crucial roles in the formation of Chinese rice wine, including the synthesis of many flavor, texture, and color metabolites (Englezos et al., 2022; Huang et al., 2019a). Krona analysis (refer to Fig. 3) revealed that the microbial communities differed significantly at different fermentation stages, which may be closely related to changes in nutrients, pH, temperature, and other biotic and abiotic factors in the fermentation environment. At the early stage of fermentation (AG0), bacteria were the dominant microorganisms and were dominated by Gammaproteobacteria and Bacilli. The high proportion of Gammaproteobacteria may be related to their greater metabolic activity in the sugar-rich environment. After 24 h of fermentation (AG24), fungi begin to dominate, especially the yeast group C. lusitaniae (refer to Figs. 4, 5, and 8), began to dominate, showing high adaptability to the environment and efficient conversion of substrates during fermentation. The rapid growth of the yeast species may be related to their ability to grow under low-oxygen or anaerobic conditions, which are common during hermetic fermentation. After 36 h (AG36), a decrease in the proportion of fungi and an increase in the proportion of bacteria were observed, which may be attributed to a reduction in fungal activity due to the depletion of available sugars in the fermentation substrate, while the bacteria adapted to this change and began to utilize the products of fungal metabolism or other nonsugar organic acids. After 43 h (AG43), Rhizopus became dominant (refer to Fig. 8), and its growth may have been due to increased acidity and the enhanced availability of certain nutrients (e.g., proteins and fats) later in the fermentation process. This change reflects the adaptive strategy of the microbial community to survive and thrive under nutrient competition and environmental stresses during fermentation.

Pediococcus is widely distributed in Jiuqu, which helps improve food taste and nutrition. The occurrence of Pediococcus pentosaceus in the AG0 samples (refer to Fig. 4) agrees with the results of a previous study (Liang et al., 2020). The presence of P. pentosaceus improved the flavor of fermented food by increasing the levels of short-chain fatty acids (SCFAs) (Jiang et al., 2021). In samples AG24 and AG36 (refer to Fig. 4), the presence of C. lusitaniae and W. anomalus can lead to the decomposition of sugar into alcohol, carbon dioxide, and other secondary products, which have far-reaching effects on the flavor and aroma of Chinese rice wine (Zhang et al., 2022b). In AG43h, Rhizopus became dominant (refer to Fig. 4) because Rhizopus can utilize macromolecular ingredients in raw materials and consume nutrients that promote the growth and reproduction of the strain (Wu et al., 2022). From the above discussion, it can be seen that the species composition of the strains in each fermentation stage is different, and the species composition is more varied. The species diversity is not the same. This result is consistent with the α-diversity analysis and β-diversity analysis results (refer to Figs. 6 and 7).

In this study, the trends in metabolic pathway changes during OFSRW fermentation were analyzed in depth, which may be closely related to the dynamics of microbial communities and metabolic activities. By comparing the abundance of metabolic pathways at different fermentation stages, this study revealed the importance and change patterns of specific metabolic pathways during fermentation. First, the abundance of organismal systemic metabolic pathways was generally greater in the AG24, AG36, and AG43 samples during fermentation than in the initial fermentation stage (AG0) (refer to Fig. 9), which may reflect the mechanism by which fermenting microorganisms respond to environmental stresses. The activation of these metabolic pathways may be related to the microbial response to oxidative stress, nutrient limitation, and other biotic stress conditions in the fermentation environment. In particular, the activation of these pathways may be related to cellular protective mechanisms such as antioxidant and damage repair functions. Second, the gradual decline in metabolism pathways with increasing fermentation time suggested that as the fermentation process proceeded (refer to Fig. 9), nutrient sources such as carbohydrates initially utilized were gradually depleted. The microorganisms had to adjust their metabolic strategies to adapt to less nutritious environments. Finally, functional analysis revealed significant carbohydrate transport, lipid transport and metabolism, and amino acid transport and metabolism pathway activity in the fermentation broth of AG0, implying that flavor formation in these samples was likely linked to protein and starch metabolism (Xiao et al., 2021; Chen et al., 2020), thus contributing to the increase in sugar content and detectable alcohol content during the middle and late stages of fermentation until the end of fermentation. These findings suggest that carbohydrate metabolism is the main pathway for microbial growth and energy production during the early stages of fermentation. The energy production of microbial flora depends mainly on the phosphorylation of substrates through sugar fermentation to acetate. However, energy production and conversion can drive the energy demand of bacterial flora, which explains the increase in bacteria in this AG24 sample. The significant activation of the metabolic pathway of energy production and conversion in the fermentation broths of AG24 suggested that C. lusitaniae and R. delemar are active during this fermentation phase and thus contribute to the increase in sugar level and acidity. The significant activation of posttranslational modifications, protein turnover, chaperones in AG36 fermentation broth, and signal transduction mechanisms in AG43 fermentation broth suggested enhanced microbially driven nutrient-seeking activity in samples from this phase (refer to Fig. 10), which may have increased microbial signal perception through a complex signaling network, enabling better nutrient utilization (Zhao et al., 2022a), especially in the case of W. anomalus and R. delemar, and significant activity (refer to Fig. 5) may contribute to the increase in alcohol and sugar levels through posttranslational modifications, protein turnover, chaperones and signal transduction mechanisms. The high expression of these metabolic pathways may, therefore, be inextricably linked to increased sugar, acidity, and alcohol contents, suggesting that some of the differences in the functional metabolic abundance of the samples from different periods may be related to the composition of the microbial community at various periods, which may help to observe changes in the volatile compound profiles between these samples.

Conclusion

This is the first report on using high-throughput sequencing to study the predicted dynamic changes in microbial communities and metabolic pathways in OFSRW natural fermentation broth. A comparison of the strains under natural fermentation conditions at different time points revealed that fungi were the dominant microorganisms during fermentation, with the number of strains detected increasing and then decreasing. However, Rhizopus was the main fungal genera present throughout the fermentation process. The increase in acidity and total sugars with increasing fermentation time was associated with L. plantarum and Mucor, and the alcohol content was not detected until the end of fermentation because Saccharomyces and Mucor were inhibited by L. plantarum during fermentation. The results of the diversity analyses revealed that the species composition of the samples varied widely and then decreased, then increased, and then declined. Energy production and conversion, carbohydrate transport, amino acid transport, and metabolism are the most active metabolic pathways in fermenters. The results of this study not only provide insights into how microorganisms adapt to environmental changes during fermentation but can also be crucial for optimizing fermentation conditions and improving product quality and flavor. Future research should focus on investigating the effects of nutrients in osmanthus petals on the flavor metabolites of sweet rice wine to increase the insights of OFSRW in flavor metabolites, which will further improve the quality and flavor of OFSRW products and enhance the status of OFSRW.

Supplemental Information

Supplemental Information 1 Function Channel Statistics.

Additional Information and Declarations

Competing Interests

The authors declare that they have no known competing financial interests or personal relationships that could have appeared to influence the work reported in this article. Ren Hongbing is the chairperson of Honghe Hongbin Foods Co., Ltd. Liu Li is employed by Honghe Hongbin Foods Co., Ltd.

Author Contributions

Ping Tian conceived and designed the experiments, performed the experiments, analyzed the data, prepared figures and/or tables, authored or reviewed drafts of the article, and approved the final draft.

Jiaqiong Wan performed the experiments, authored or reviewed drafts of the article, and approved the final draft.

Tuo Yin analyzed the data, authored or reviewed drafts of the article, and approved the final draft.

Li Liu performed the experiments, prepared figures and/or tables, providing and experimental equipment, guiding the production process, and approved the final draft.

Hongbing Ren conceived and designed the experiments, prepared figures and/or tables, providing production site and experimental equipment, funding acquisition, and approved the final draft.

Hanbing Cai analyzed the data, prepared figures and/or tables, and approved the final draft.

Xiaozhen Liu analyzed the data, authored or reviewed drafts of the article, and approved the final draft.

Hanyao Zhang conceived and designed the experiments, authored or reviewed drafts of the article, and approved the final draft.

Data Availability

The following information was supplied regarding data availability:

The data is available at NCBI SRA: PRJNA1131309.

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
