# Peer review of "Acidity, sugar, and alcohol contents during the fermentation of Osmanthus-flavored sweet rice wine and microbial community dynamics"

_PeerJ, doi:10.7717/peerj.18826_

## Round 0.1 · original submission · Major Revisions

Thank you authors for your patience as reviewers attended to your scholarly contribution. You can see that several concerns have been raised. Please attend to them diligently. Ensure to provide a detailed response in the revised manuscript, as well as to the comments raised.

Reviewer 1 ·

Basic reporting

This study exploited the Acidity, sugar, and alcohol during the fermentation of Osmanthus-flavored sweet rice wine and microbial community dynamics. Overall, this paper is adequately written with sufficient results/data and may be accepted for publication after revision based on the suggested amendments as following.

Experimental design

1. Line 30 (abstract): “cinnamon-flavored sweet rice wine”: briefly state the rationale behind the use of cinnamon.
2. Line 155: it was unclear the use of 0.4% wine in the sample preparation, please justify.
3. Line 159-160: how was the fermentation mash was collected prior to high-throughput sequencing analysis? Method for microbial sample for total DNA extraction can be more detailed

Validity of the findings

4. Line 314-317: “Rhizopus has abundant amylase and certain liquefaction enzymes, which can chain OFSRW with saccharification and fermentation throughout the whole fermentation process from the beginning to the end of the fermentation process, and the fermentation effect was more thorough; thus, the starch yield further improved”. Please add supporting references and explain clearer how the fermentation effect was more thorough with starch yield further improved.
5. Figures 3, 5 and 8 need to be enlarged for better reading.

Additional comments

no comment

·

Basic reporting

1. While the authors have provided a comprehensive background of the topic, the complexity of some sentences has made comprehension difficult. I suggest that the authors should revise sentences in lines 50 – 56.
Literature references have been provided, covering the research's background.
The article is also well-structured. However, the authors need to do the following to improve this section;
6. Table S1 is mentioned in the text (line 139) but is not found in the current work.
7. Present the values of the physicochemical properties in Table 1 in mean ± standard deviation
8. Present all figure titles and legends below the corresponding figure and not above the statistics, as is the case with the current manuscript.
9. The quality of Figure five should be improved as some vital information in the figure is not visible

Experimental design

2. The research objectives should be clearly stated in simple terms in lines 125 to 139. I propose that the authors use less than five lines for this.
3. Consider taking the information presented in lines 130 to 139 to the materials and methods section
4. The authors have provided comprehensive methods for addressing the research questions. However, the section describing the processing of OFSRW (lines 143 to 160) can be made more comprehensible by presenting it as a flowchart and inserting the processing conditions in the chart.
5. I also recommend that the authors provide the experimental design and the basis for accepting or rejecting the hypothesis (p-value).

Validity of the findings

The findings are novel, sound and valid. However, the conclusion can be improved by simply stating the main findings and not necessarily repeating the discussion or stating reasons for the behaviour of the results. E.G., lines 524 to 528

Additional comments

The authors have done great work studying the dynamics of the quality changes in the production of traditional Chinese rice wine that the Chinese people widely love. The background is well-covered with a clear objective. The materials and methods are comprehensive enough to address the goals, and the results align with the other results in the literature. The conclusion is also in line with the objectives. However, there are minor corrections annotated in the pdf file and also presented in the reviewer’s comment file to improve the overall quality of the manuscript.

·

Basic reporting

The authors of this manuscript presented scientific results on an important issue of microbial quality analysis of traditional Chinese rice wine. They presented the manuscript as:
Unambiguous, professional English is used throughout the manuscript.
The literature is well-referenced, and sufficient field background/context is provided.
Professional article structure, figures, tables. Raw data shared.
Self-contained with relevant results to hypotheses.

Experimental design

The experimental design is well organized.
Original primary research within Aims and Scope of the journal is presented.
The research question well defined, relevant & meaningful. It is stated how research fills an identified knowledge gap.
Rigorous investigation performed to a high technical & ethical standard.
Methods described with sufficient detail & information to replicate.
However, there are some information missed such as:
Whose methods/standard procedures are the stated determination of Brix, Acidity, and Alcohol Content (needs citation)? If these were used for the first time optimized experimental conditions are required.
How sugar content can be determined using direct titration method?

Validity of the findings

Meaningful replication is encouraged where rationale & benefit to literature is clearly stated.
All underlying data have been provided; they are robust, statistically sound, & controlled.
Conclusions are well stated, linked to the original research question & limited to supporting results.

Reviewer 4 ·

Basic reporting

The manuscript presents an interesting study that explores the dynamics of sugar levels, acidity, alcohol content, and microbial community during the fermentation process of Osmanthus-flavored sweet rice wine (OFSRW). The study is valuable, as it offers insights into the microbial composition and metabolic pathways that impact fermentation quality. However, several issues need to be addressed for clarity and precision

Experimental design

1) The description of the methodology is generally clear, but there are some sections where the flow of information can be improved for better readability. The transitions between processes (e.g., from soaking to steaming, and then from cooling to fermentation)
2) Line 143-144: The description of "Angie's Sweet Wine Quartz" should clarify whether this is a commercially available yeast product, a specific strain, or something else. If it’s a proprietary product, a reference or further clarification is needed.
3) Line 146-147 : The use of arrows in the article should be replaced with written text
4) Line 154: The description of the desired texture of the rice (“hard and soft outside and inside, loose but not rotten”) is qualitative. If possible, providing more objective measurements would enhance reproducibility
5) Line 157: The term “compacted” in the context of fermentation is ambiguous. Does it refer to physical compression to remove air, or another form of handling? Clarifying this will help other researchers replicate the method
6) Line 164-165: For the acidity and alcohol content measurements, It is recommended to either describe the method in detail or include the relevant procedural standards within the manuscript.
7) Line 167: The CTAB method for DNA extraction is appropriate, but the specific protocol (e.g., concentrations, incubation times) should be provided or referenced in detail. Also, whether any modifications were made to Liu et al. (2017)'s method should be stated
8) Line 241-243: A mention of how the bar charts for metabolic pathway analysis were validated or tested for statistical significance is missing

Validity of the findings

1) Line 250-251: The suggestion that low acidity may be related to the activity of L. plantarum is interesting, but more data are required to support this claim. Was microbial analysis conducted at this stage to detect the presence of L. plantarum or other microorganisms? Additionally, the correlation between acidity and L. plantarum activity should be supported by references to previous studies or empirical data from this study
2) Line 251-254: The manuscript should provide more detailed speculation on why this delay in alcohol production occurs. Discussing possible reasons for the absence of alcohol at mid-fermentation would strengthen this section
3) Line 258-259: This statement is sound but could benefit from the addition of supporting references that explain how acidity modulates microbial populations, specifically in traditional rice wine fermentations
4) Line 259-261: The manuscript should clarify whether a 2% alcohol content is typical for this stage of osmanthus flower sweet rice wine fermentation. In most alcoholic fermentations, one would expect a higher alcohol concentration after 43 hours. If this is an expected result for this type of fermentation, briefly explaining the unique characteristics of OFSRW that contribute to the lower alcohol content would enhance the reader's understanding
5) Line 261-263: The suggestion that L. plantarum may inhibit Saccharomyces and Rhizopus activity is interesting, but more evidence is needed to support this hypothesis. Did you observe population changes in Saccharomyces and Rhizopus during this time? Additionally, could other microbial species or fermentation conditions contribute to this inhibition? If possible, present supporting data or references for this claim

Additional comments

The manuscript presents an insightful study investigating the changes occurring during the fermentation of Osmanthus-flavored sweet rice wine (OFSRW). This research holds significant potential for enhancing the value of local products

---

## Round 0.2 · Minor Revisions

Please, authors , you can see the reviewers very much appreciate your revised work.

A few minor comments were raised. Kindly address the comments.

I look forward to your revised manuscript
Thank you

·

Basic reporting

The authors have done a great job addressing the reviewer's comments, and the manuscript has been dramatically improved. However,
Lines 202 to 204 should be reported in the past tense to show what you did and the quantities of the supernatant volumes you added to the collection tubes, not ≤ or approximately as you reported in the methodology.
In lines 133 to 134, delete “in this work, was used.”
Delete “was as follows” in line 137
In line 301, Wang et al., 2013 was mentioned in the text, but it was not included in the reference list.
In line 785, under the reference list, Wang XY, Wu MN, Yu QR, Ma LX, Yao D, Zhang LY. 2023. Isolation and Identification of Lactiplantibacillus plantarum ST3.5 and Its Inhibitory Effect on Mold [J]. Science and Technology of Food Industry, 44(13): 141−149. (in Chinese with English abstract). DOI: 10.13386/j.issn1002-0306.2022080334 was mentioned. Kindly check and correct

Experimental design

The experimental design has been well captured

Validity of the findings

The findings are valid

·

Basic reporting

No comment

Experimental design

No comment

Validity of the findings

No comment

Additional comments

No comment

Reviewer 4 ·

Basic reporting

The author has revised according to the suggestions; therefore, no further recommendations are needed

Experimental design

The author has revised according to the suggestions; therefore, no further recommendations are needed.

Validity of the findings

The author has revised according to the suggestions; therefore, no further recommendations are needed.

Additional comments

The author has revised according to the suggestions; therefore, no further recommendations are needed.

---

## Round 0.3 · Minor Revisions

Thank you authors for your patience as reviewers checked your revised manuscript. The reviewers are positive about your revision. However, some areas need your careful attention:

a) Introduction: Kindly find the reference(s) to support the claim made in lines 91-96, and 97 to 103.

Paragraph 3 talks about OFSRW fermentation, and introduces traditional aspects, and techniques to assess it, but does not clearly show the gap that this current work is trying to fill. Provide two sentences, starting from line 127 to begin the 4th paragraph telling us what the gap is, before stating the objective. Also, kindly touch lines 127 to 135, it is not connecting each other, there appears to be some lack of flow in reading

b) In the methods, please provide a schematic flow to support the description of high-throughput sequencing, and refer to this figure in that specific section, from solution C1 to C6.

Although you used reference of Rognes et al 2016 for sequence processing, please elaborate on it, succinctly, the key aspects, this is important to guide readers. Same applies to analysis of changes in species composition, and alpha, beta diversity analysis.

The editor observes various statistical methods were employed, it is important that authors carefully identify all the statistical analyses aspects, and still mention them again in a separate 'Statistical analyses" section that would end the materials and methods. Begin this section by, "For emphasis, the statistical analyses employed as abovementioned included....(then enumerate them again, and tell us why each is employed)

c) Results are well detailed, however, given the results and separated from discussion, please authors, kindly make sure that all figures mentioned in the results section, are all captured in the discussion. Use (Refer to Figure x) to capture all the data of specific figures being discussed. Make sure that all the figures are captured. Please, this is compulsory

Also, in your discussion, strengthen your discussion on the how? and why? Why is the trends observed, and how does the literature explain it in the context of your study. Tell us much less about what the literature said.

d) In your conclusion, kindly provide direction for future studies, based on the totality of your findings.

Look forward to your revised manuscript.

·

Basic reporting

Satisfactory

Experimental design

Satisfactory

Validity of the findings

Findings are valid

Additional comments

Satisfactory

---

## Round 0.4 · Minor Revisions

Thank you authors for your revised manuscript.
After a careful and meticulous check, many of the queries were not addressed
Kindly address all concerns raised carefully and completely
Thank you


Thank you authors for your patience as reviewers checked your revised manuscript. The reviewers are positive about your revision. However, some areas need your careful attention:

a) Introduction: Kindly find the reference(s) to support the claim made in lines 91-96, and 97 to 103.

Paragraph 3 talks about OFSRW fermentation, and introduces traditional aspects, and techniques to assess it, but does not clearly show the gap that this current work is trying to fill. Provide two sentences, starting from line 127 to begin the 4th paragraph telling us what the gap is, before stating the objective. Also, kindly touch lines 127 to 135, it is not connecting each other, there appears to be some lack of flow in reading

b) In the methods, please provide a schematic flow to support the description of high-throughput sequencing, and refer to this figure in that specific section, from solution C1 to C6.

Although you used reference of Rognes et al 2016 for sequence processing, please elaborate on it, succinctly, the key aspects, this is important to guide readers. Same applies to analysis of changes in species composition, and alpha, beta diversity analysis.

The editor observes various statistical methods were employed, it is important that authors carefully identify all the statistical analyses aspects, and still mention them again in a separate 'Statistical analyses" section that would end the materials and methods. Begin this section by, "For emphasis, the statistical analyses employed as abovementioned included....(then enumerate them again, and tell us why each is employed)

c) Results are well detailed, however, given the results and separated from discussion, please authors, kindly make sure that all figures mentioned in the results section, are all captured in the discussion. Use (Refer to Figure x) to capture all the data of specific figures being discussed. Make sure that all the figures are captured. Please, this is compulsory

Also, in your discussion, strengthen your discussion on the how? and why? Why is the trends observed, and how does the literature explain it in the context of your study. Tell us much less about what the literature said.

d) In your conclusion, kindly provide direction for future studies, based on the totality of your findings.

Look forward to your revised manuscript.

---

## Round 0.5 · accepted · Accept

Authors, thank you very much for addressing all concerns, and revising your work to a very high level. It is now acceptable for publication. Very grateful for your great scholarly work, and genuinely hope for your continued future contributions to PeerJ.